# Administration of Defective Virus Inhibits Dengue Transmission into Mosquitoes

**DOI:** 10.3390/v12050558

**Published:** 2020-05-18

**Authors:** Tarunendu Mapder, John Aaskov, Kevin Burrage

**Affiliations:** 1School of Mathematical Sciences, Queensland University of Technology, Brisbane, QLD 4000, Australia; kevin.burrage@qut.edu.au; 2Australian Research Council Centre of Excellence for Mathematical and Statistical Frontiers, Queensland University of Technology, Brisbane, QLD 4000, Australia; 3Institute of Health and Biomedical Innovation, Queensland University of Technology, Brisbane, QLD 4059, Australia; j.aaskov@qut.edu.au; 4Department of Computer Science, University of Oxford, Oxford OX1 3QD, UK

**Keywords:** defective particles, dengue virus, dengue transmission, population of models, mosquito infectious dose

## Abstract

The host-vector shuttle and the bottleneck in dengue transmission is a significant aspect with regard to the study of dengue outbreaks. As mosquitoes require 100–1000 times more virus to become infected than human, the transmission of dengue virus from human to mosquito is a vulnerability that can be targeted to improve disease control. In order to capture the heterogeneity in the infectiousness of an infected patient population towards the mosquito population, we calibrate a population of host-to-vector virus transmission models based on an experimentally quantified infected fraction of a mosquito population. Once the population of models is well-calibrated, we deploy a population of controls that helps to inhibit the human-to-mosquito transmission of the dengue virus indirectly by reducing the viral load in the patient body fluid. We use an optimal bang-bang control on the administration of the defective virus (transmissible interfering particles (TIPs)) to symptomatic patients in the course of their febrile period and observe the dynamics in successful reduction of dengue spread into mosquitoes.

## 1. Introduction

Dengue is a major health burden in tropical regions [1,2], which is caused by the systematic and self-limited transmission of dengue virus (DENV) between human and mosquitoes [3]. There are four serotypes of the DENV arbovirus which have been identified as being transmitted by *Aedis aegypti* mosquitoes [4]. As *Aedis aegypti* is a competent vector for all dengue serotypes, DENV can systematically infect and persist within the mosquito body, and eventually the vector becomes infectious [5,6,7,8]. The presence of DENV in the abdomen or saliva defines a mosquito to be infected or infectious, respectively. However, the passage of DENV from the mosquito mid-gut to the salivary gland needs to cross a number of tissue barriers and innate immunity responses. To cross these hurdles, DENV needs to replicate across all the different tissues and body fluids [9,10]. For this reason, a bottleneck arises during the transmission flow from the human host to the mosquitoes. The mechanisms underlying the transmission of virus from human to mosquito can be exploited as a potential effective dengue control.

Traditional vector controls based on reducing the mosquito population have not been successful in endemic and pandemic situations for dengue [11]. Destroying mosquito habitats using insecticides and biological controls by nematodes and fungi are primitive approaches for vector control. Due to the lack of success of these approaches [12,13], genetic manipulation and population replacement of mosquitoes has recently been attempted [3,14,15]. This has not been successful due to the inability of these populations to invade the wild mosquito population. On the other hand, the global spread of dengue is accentuated through increased travel between countries and urban growth [16,17].

Another popular control of infectious disease spread is through vaccination. However, the most crucial reason for the failure of the conventional vaccination programs for RNA viral diseases is the frequent mutation and fast evolution rate. The main reason is that developing a vaccine for dengue is difficult because of cross-reactivity between serotypes. Currently, many vaccines (including the most popular Dengvaxia^®^) for dengue virus are in development and clinical trial. Most of them use live attenuated virus [18], inactivated virus [19], recombinant subunit vaccines [20], delivery vectors [21] or DNA plasmids [22] but none of them deploy defective interfering (DI) particles and their transmission between the host and the vector. Thus, an efficient vaccination strategy should be co-evolving and persist over multiple passages and through the host-vector transmission shuttle.

Defective interfering (DI) viruses, containing nucleotide deletions, can be deployed as a potential co-evolving intervention. In the case of RNA viruses, defective viral genomes (DVGs) are spontaneously generated within infected host cells [23,24]. Upon infection, the (+)ssRNA appears as mRNA for translation of the replication and encapsidation machinery. The dynamic secondary and tertiary structures of (+)ssRNA plays a crucial role in regulation of translation, transcription, replication and encapsidation [25]. Due to the RNA structure-mediated binding of the polymerase, ‘snapback’ and ‘copyback’ replication mechanisms can generate defective genomes [26]. Within infected host cells, the defective RNAs interfere with virus replication and help attenuate the plasma viraemia level in the host body [27,28]. DI particles are so named because of the interference with the multiplication of standard virus particles. DI particles and standard virus stimulate the host immune responses in the same way as the DI particles are encapsidated by the normal coat proteins synthesised by the standard virus [29]. There is a large number of models studied on the generation and activities of DI particles for many animal viruses [30,31].

We have recently proposed a within-host model of dengue virus infection that covers the natural synthesis of dengue DI particles [32]. We have constructed a population of models (POMs), calibrated with clinical data for 208 patients [33]. The intention of the POMs study is to capture the biological heterogeneity in, for example, patient response, which is difficult to investigate with experimental methods alone [34]. One cause of variability is the cellular infection responsible for the viral load in the patients’ blood samples. Infection and super-infection by standard virus and DI particles introduce variability at numerous stages, including the membrane receptor dynamics of early-infected cells, and activation of the interferon pathways in the late-infected cells [35,36]. A total of four clinically calibrated serotype-specific POMs have been employed to investigate variability in dengue infection in physiological and pathological conditions. In the same framework, we have discussed the effect of adding excess DI particles through a bang-bang optimal control as an intervention strategy. Addition of excess DI particles to a host system that naturally generates DI particles, helps to reduce plasma viraemia peak and duration. Hence, we have predicted four serotype-specific populations of DI particles-mediated controls. We identify the population of vectors of doses and durations of DI particles addition as population of controls (POCs). Naturally, we observe attenuation of the viraemia peaks and duration after successful deployment of the POCs and now the POMs are known as the controlled POMs (cPOMs). The POCs have also been characterised on the basis of its efficiency in reduction of the within-host viral burdens.

The transmission of dengue virus into mosquitoes is directly proportional to the viral load of the infected patients and the infectiousness of the patients varies with the days of febrile period. In this paper, we elucidate the dynamics of host-to-vector transmission using the outcomes of our previous model [32]. The profiles of virus and DI particles from the within-host model are implemented as the infecting inputs to the mosquito transmission model. We calibrate the population of transmission models with the mosquito infection data from the exposure experiments. Our hypothesis is that controlling the plasma viraemia within the infected host may efficiently restrict the host-to-vector transmission and hence the outbreak of dengue. Therefore, the controlled viraemia is further used in the transmission model to observe reduced infection in the mosquito population.

## 2. Methods

The key aim of this paper is to study the transmission dynamics of DENV from human host to vector (mosquito) and the effect of adding excess DI particles on the virus transmission. To validate and calibrate the models, we utilized a set of clinical data for 208 adult dengue patients, collected by Nguyen et. al. in Vietnam [33]. They recorded the plasma serology profiles and quantified viraemia levels on a daily basis. In a previous study, we have deployed the patients’ blood sample data to calibrate a population of within-host models (POMs) and predicted a population of personalised controls (POCs) that is able to reduce blood viraemia levels [32]. Nguyen et. al. also performed a mosquito exposure experiment on the same group of patients. With written informed consent, each patient was assigned for multiple exposures to mature and competent mosquitoes. 408 exposures were scheduled purely randomly during high fever days. In each exposure, approximately 25–40 female 3–7 day old *Aedis aegypti* mosquitoes were allowed to bite on the patient’s forearm. The successful human-to-mosquito transmission of DENV was estimated by RT-PCR (reverse transcriptase polymerase chain reaction) of the viral titre from the mosquito abdomens [33]. To correlate the patients’ plasma viraemia kinetics with the mosquito infection pattern, the blood-fed mosquitoes were harvested 12 days after the day of exposure. The efficiency of patient viraemia to infect the mosquito population was measured by the mosquito infectious dose (MID) in a quantitative way. The 50% mosquito infectious dose (${\mathrm{MID}}_{50}$) is a direct evaluation of the force of host-to-vector virus transmission. It can be estimated by the comparison of the DENV plasma viral load during the exposure experiment with the proportion of the DENV-infected blood-fed mosquito cohort. We used the dataset of mosquito exposures to help us explore the host-to-vector transmission dynamics. In this paper, we modeled the mosquito exposure experiment to occur in an open environment within the territory of a mosquito population. Hence, an experimentally calibrated population of transmission models was proposed for each dengue serotype. Here we note that the serotype-specific POMs were calibrated only based on the recorded dataset, otherwise the models followed the same mathematical structure. We were unable to incorporate any serotype-specific interaction based on the available data of viral load in the mosquito abdomen and patients’ serology.

### 2.1. Within-Host Dengue Viraemia

The data of virus and DI particles in the patients’ blood samples were used as input for the present model. The kinetics of plasma viraemia were estimated from a within-host dengue virus infection model [32]. The within-host model described the dynamics of cell-virus interaction and the triggered adaptive immune response inside the human host body. As the DI particles could replicate only in the co-infected cells, these cells were the only source in which to accumulate DI particles in the host body fluid. Hence, every time an infected host with high plasma viraemia was exposed to mosquitoes, there was a possibility that the mosquitoes while taking a blood meal were infected in three different ways: by the virus, by the DI particles or dually. In the within-host model we also predicted optimal bang-bang control (see [32] and discussed later) with excess DI particles addition leading to patients’ viraemia reduction. Hence, the controlled viraemia profiles are available from the within-host model [32] to use in the present model to observe their efficiency in transmission reduction.

### 2.2. Dengue Transmission in Mosquitoes

With the natural birth-death flow of mosquitoes this model examines how a population of mosquitoes take blood meal from infected patients on different days of their febrile period in a continuous manner (Figure 1). The reported data were collected on specific days, on which the mosquitoes are harvested for quantitative assays. Hence, we found a daily distribution of infected mosquitoes in order to calibrate our population of models. In general, many mathematical models of infectious disease spread follow the very well-established SIR model [37,38]. Variants include adding compartmental models, seasonal effects and spatio-temporal dynamics [38,39,40]. However, tracking the transmission of the host plasma viral load to the mosquito population within the febrile period has not yet been explored. We followed the traditional SIR approach but did not include the recovered (R) compartment, as mosquitoes die naturally before they can recover from dengue virus infection [41]. It is possible to derive a stochastic model to address the underlying fluctuations, but our approach was conceived in a deterministic ordinary differential equation setting as the high levels of interacting viraemia produces meanfield approximation as a continuum. Moreover, we adopted the method of population of models to characterize the inherent variability. We constructed a population of transmission model that utilized the patient viraemia profiles from our previous model [32]. This model can explain the infected fraction of the mosquito population in terms of the transmission of viraemia from the infectious patients. The mathematical structure of the model is shown below.
(1)$$\begin{array}{cc} \frac{dS}{dt} & =A-(\mu +\phi (V(t)+D(t)))S\\ \frac{dI_{V}}{dt} & =\phi SV\left(t\right)-\mu I_{V}\\ \frac{dI_{D}}{dt} & =\phi SD\left(t\right)-\mu I_{D}\\ \frac{dI_{VD}}{dt} & =\phi \left(D\left(t\right)I_{V}+V\left(t\right)I_{D}\right)-\mu I_{VD}. \end{array}$$

Here, *S*, $I_{V}$, $I_{D}$, and $I_{VD}$ are the susceptible mosquitoes and three kinds of infected pools of mature mosquitoes, respectively. The infected pools are by virus only ($I_{V}$), by DI particles only ($I_{D}$) and by both virus and DI particles ($I_{VD}$). *A*, ϕ and μ are the rate of reproduction, infection and natural death of mosquitoes, respectively. The plasma viral load (V(t)) and DI particles (D(t)) are obtained from our previous study [32]. We denote each fraction of the infected mosquito population with respect to the entire mosquito population as
(2)$$\begin{array}{cc} {\tilde{I}}_{V} & =\frac{I_{V}}{\left(S+I_{V}+I_{D}+I_{VD}\right)}\\ {\tilde{I}}_{D} & =\frac{I_{D}}{\left(S+I_{V}+I_{D}+I_{VD}\right)}\\ {\tilde{I}}_{VD} & =\frac{I_{VD}}{\left(S+I_{V}+I_{D}+I_{VD}\right)}. \end{array}$$

### 2.3. Population of Models

Characterizing the variability between individuals from the same species is fundamental in biology. In our setting, every infected individual in a population may not produce similar infectiousness to the similar groups of susceptible, mature mosquitoes. Two patients with similar viraemia profiles, for example, when exposed to the same mosquito populations may show very different infectiousness. In such cases, the use of a variable population of models is more realistic than a single mean-field model [42,43]. The underlying variability may be manifested by: (1) the patients’ physiology and immunological response, (2) behaviour of the uptaken virus on interaction with the host cells, (3) mosquito physiology. The first two points were covered when we used a calibrated population of patients’ viral load (V(t)) and associated DI particles (D(t)). To capture the variability in patient infectiousness towards mosquitoes, a population of models approach was used for the transmission model. The main purpose of such a study was to decipher how a particular population of patients was different from the others (i.e., infectiousness to mosquitoes at different levels of viral load) and on the clinical realm, compared with the treated (controlled) population. In a clinical and experimental framework, it is difficult to track the variability, so a population of mathematical models can overcome this problem even in a low sample size.

The population of transmission models takes the form in Equation  (Equation 1) and is calibrated by sampling the model parameters based on Latin Hypercube Sampling (LHS) [44]. Uniformly selected 15% profiles of viraemia (V(t)) and DI particles (D(t)) from the infected patients’ population are used here as model input and the parameters (*A*, μ and ϕ) are sampled from LHS. We are free to set different criteria for our model calibration. A previously reported article has chosen the range of the biomarker data as the calibration criteria [42], which is sometimes coarse. In our recent article we have calibrated models by matching the distribution of the data [45]. As distribution matching is computationally expensive and an initial observation did not offer significant insight in our present study, we adopt the previous range-based method as described in Algorithm 1. The model calibration process is based on the range of the experimental biomarker data [32,34]. LHS is a sampling technique for high dimensional parameter space so that each sample is the only one from each axis-aligned hyperplane containing it. The advantage of LHS over uniform random sampling is higher coverage of the high dimensional parameter space, no scaling with dimension and a form of variance reduction [44]. In the present paper, for each input patient’s model with a particular serotype, we simulated the same model (Equation (Equation 1)) for 1000 sampled sets of parameters from a three dimensional LHS and selected only those models that appeared with ${\tilde{I}}_{V}$ levels within the ranges of the data. The allowed ranges of the parameters were: $A\to (5.0\times 10^{-3}-50.0)$ per day, $\mu \to (1.567\times 10^{-4}-1.567)$ per day and $\phi \to (7.5\times 10^{-11}-7.5\times 10^{-7})$ per day, which were within the biologically feasible ranges [38]. The models that produced the daily infected mosquito population within the range of reported data were included in the population from the initial population of 1000 sampled models for each individual patient model. We constructed four separate populations of the different fractions of infected mosquito population for four serotypes of DENV.
**Algorithm 1** Construction of experimentally calibrated transmission POMs 1: Nsample←numberofsamples 2: Npat←numberofwithin-hostmodelsinthepopulation 3: Nparam←numberofmodelparameters**Require:**  mossydata←**Read** human-to-mosquito transmission data files 4: figure←**Draw**
mossydata outputs 5: **for**
i←1,Npat
**do**
**Require:**  hostmodel←**Read** within-host model data files 6:  t,V,D←viraemiadynamics 7:  param← Perform LHS and build (Nsample×Nparam)-dimensional parameter hyperspace 8:  **for**
j←1,Nsample
**do**
 9:   mossymodel(j)←**Solve** the model for param(j)10:   **if**
mossymodel(j)≤ range of mossydata(j)
**then**11:     **Accept**
param(j) into POMs12:     figure←**Draw**
mossymodel(j) outputs13:
  **goto**
*8*
14: **goto***5*15: **close**;

### 2.4. Optimal Bang-Bang Control

In the present study, we used the output from the same population of within-host models twice; first, the uncontrolled V(t) and D(t) while constructing the infection population of mosquitoes and secondly, the controlled V(t) and D(t), which led to high reduction of the host-vector transmission of dengue virus. Here the controlled V(t) and D(t) appeared after application of an optimal bang-bang control in the within-host POMs [32]. The purpose of optimal control was to determine the optimal trajectory of a control variable over time by optimizing a predefined objective function using dynamical programming. Pontryagin’s minimum principle and Hamilton-Jacobi-Bellman Theorem are popularly used to solve dynamical control problems [46,47]. In dynamical programming, optimal controls are of two kinds: continuous and bang-bang. Although continuous optimal control is popular in engineering and biology, bang-bang control is less popular due to computational difficulties. Bang-bang control flips between the ‘on’ and ‘off’ states depending on the states of the system; hence it is more relevant in clinical interventions. Recently, we have discussed the successful implementation of both continuous and bang-bang optimal control to predict interventions in disease models such as acute myeloid leukaemia and dengue infection [32,48].

### 2.5. Jensen-Shannon Divergence

As we considered population of all plausible mosquito transmission models for each patient viraemia profile, the mosquito infection fractions (${\tilde{I}}_{V}$ and ${\tilde{I}}_{VC}$) were time-dependent distributions for each value of V(t) on each day. Here we denoted the controlled ${\tilde{I}}_{V}$ as ${\tilde{I}}_{VC}$. Once the V(t) was substituted by the controlled V(t) for a patient model, the constructed POMs was also substituted by the cPOMs and ${\tilde{I}}_{V}$ by ${\tilde{I}}_{VC}$. Now the reduction in ${\tilde{I}}_{VC}$ from ${\tilde{I}}_{V}$ could not be estimated in either way but a difference or distance between two distributions. The Jensen-Shannon Divergence (JSD, D) was the most suitable for the present scenario. To estimate the efficiency of the DI particles-mediated intervention over virus transmission, we used JSD, which is derived from the Jensen’s inequality and Shannon entropy [49]. The JSD measure is a symmetric overall difference between two distributions. We calculated the JSD between the uncontrolled and controlled infected mosquito populations for each patient model on each day of illness as
(3)$$D\left({\tilde{I}}_{V},{\tilde{I}}_{VC}\right)=\frac{1}{2}\left(KL\left(P_{{\tilde{I}}_{V}},P_{{\tilde{I}}_{VC}}\right)+KL\left(P_{{\tilde{I}}_{VC}},P_{{\tilde{I}}_{V}}\right)\right),$$
where, KL is the Kullback-Leibler divergence measure [50] between the probability distributions of uncontrolled and controlled infected mosquito fractions ($P_{{\tilde{I}}_{VC}}$ and $P_{{\tilde{I}}_{V}}$) for each individual patient model.

## 3. Results

The profiles of ${\tilde{I}}_{V}$, ${\tilde{I}}_{D}$ and ${\tilde{I}}_{VD}$ are shown in Figure 2. The ${\tilde{I}}_{V}$ fraction was calibrated using the available clinical data, shown in black dots for 208 patients, assigned in 408 mosquito exposure experiments. On an average, every patient was exposed twice within day 2 and day 7 of illness. The other two panels, ${\tilde{I}}_{D}$ and ${\tilde{I}}_{VD}$, were estimated from the simulated models while calibrating the POMs. The patient-specific POMs are shown in different colours in Figure 2. These calibrated populations of transmission models were simulated again with the treated (controlled) viraemias. The treated viraemias were obtained from the corresponding within-host models after applying bang-bang optimal control. As the controlled within-host viraemias were highly reduced, the transmission of the viraemias into mosquito population was also significantly low.

In Figure 3, the selected parameters for the calibrated models are shown in box plots for the four dengue serotypes. During the calibration of POMs with the reported biomarker data, the different rate constants of mosquito reproduction (*A*), infection by host viraemia (ϕ) and natural mosquito death (μ) were sampled using LHS and selected within the POMs according to Algorithm 1. The accepted model parameters were normalised by their values and included in the box plot as scattered distributions.

In the present study the plausible ranges of MIDs were estimated from the four serotype-specific calibrated POMs (Figure 4). The infected mosquito fraction (${\tilde{I}}_{V}$) was plotted with respect to the Log10 of employed viral load (V(t)) from the within-host model [32]. The distribution of the 50% mosquito infectious dose (${\mathrm{MID}}_{50}$) was estimated for each serotype from the fraction of ${\tilde{I}}_{V50}$s and the corresponding V(t) levels. As a patient with high viraemia was maximally efficient for transmission, very few transmissions were observed after day 5 of illness.

In Figure 5 the POMs for the four serotypes (across the columns) simulated with the controlled viraemia from the within-host model as the input are presented. If we observe the infection in mosquitoes by the virus (Figure 5), the ${\tilde{I}}_{V}$ levels for all DENV serotypes were reduced more than 10-fold. For DENV-1 and DENV-2, the overall peaks of ${\tilde{I}}_{V}$ were close to 0.05, whereas the same peaks for the uncontrolled case (Figure 2) were near 0.9 and 0.85. We observed more efficient reduction in ${\tilde{I}}_{V}$ for DENV-3 and DENV-4. The controlled ${\tilde{I}}_{V}$ for DENV-3 was near 0.01 and for DENV-4 it was near 0.02. The transmission of DI infection (${\tilde{I}}_{D}$) was increased significantly for all the serotypes, but it could not dominate the transmission of ${\tilde{I}}_{VD}$. The most interesting results were observed in the case of DENV-2 and DENV-3, where a majority of the patients appeared with secondary infection. The utilization of the controlled viraemia did not greatly affect co-infection (${\tilde{I}}_{V}$) profiles during the virus infection reduction in those case of secondary infections. Although the co-infection population dynamics showed changes with respect to the population before intervention, the individual profiles in the whole population did not get affected. Notable reduction was observed in the case of DENV-1 and DENV-4 while DENV-2 and DENV-3 were still in a good state.

Figure 6 shows the controlled ranges of mosquito infectious dose (MID) for the four DENV serotypes. In most of the cases of DENV-1 and DENV-2, the ${\tilde{I}}_{V}$ reduced ≈100−fold with the reduction in viraemia. However, some controlled MID points were observed as outliers in the population of DENV-2. If we observe these outlying points carefully, we can make a note that these points appeared at the controlled peak values of the viraemias along the x-axis. The corresponding controlled ${\tilde{I}}_{V}$ fractions (in cyan, Figure 5) had similar short-lived sharp peaks between day 0 and day 1 of illness. On this point we want to mention that dengue transmission by asymptomatic patients has been reported significantly more infectious than clinically symptomatic patients in some cases [51]. However, the outlying points in our controlled ${\tilde{I}}_{V}$ fractions were not in the region of significant transmission. On the other hand, with the same reduction in viraemia for DENV-3 and DENV-4, ${\tilde{I}}_{V}$ was reduced much more efficiently and they did not possess any such outlying points for ${\tilde{I}}_{V}$ fractions (Figure 6).

In Figure 7 the box plots in the inset produce the distribution of all the patients’ performance on viraemia reduction in terms of JSD. A greater JSD score implies more effective control and with that notion, DENV-2 and DENV-3 show more effectiveness with higher JSD values. Moreover, we know that an intervention strategy using excess DI particles will be called efficiently cost-effective if it can reduce maximum transmission with a minimum DI particles addition for minimum duration. Hence, we present the distribution of JSD with respect to the normalized control expense (*C*) and find DENV-2 with the most efficient control strategy for the present dataset. We calculated the area under the control curve as the control expense for each patient model [32] and that was normalized for the population.

## 4. Discussion

The present study assumes that dengue virus transmission in mosquitoes is directly proportional to the host viraemia level and the transmission can be mitigated by controlling the host viral load. The main goal of this paper is to calibrate population of models (POMs) for the infectiousness of the patients to mosquitoes and to predict the efficiency of DI particles treatment in reduction of the transmission. The method of POMs helps to unlock the underlying variability of the data and predict the feasible regime of the model. Optimal bang-bang control suggests the efficiency of the DI particles mediated intervention strategy using the J-S divergence. Moreover, the DI particles cannot be transmitted independently; there is a trade-off between assisted DI transmission and co-infection.

It is clear from Figure 2 that as long as the host viraemia level stays very high, i.e., within 2–4 days of illness, the probability of transmission into mosquitoes is high. One of the reasons behind this phenomenon is the requirement of high viral load (see Figure 4) in the host blood for successful systematic replication through the tissue barriers in the mosquito body. Another reason is short mosquito lifespan (approximately 15 days) that sometimes terminates before the mosquito becomes systematically infected. As Figure 4 cannot interpret the relative timescales of the host plasma viraemia and mosquito infection, we have to compare the time profiles of the mosquito infections (see Figure 2) and viraemia profiles reported before [32]. The rise of the viraemia-infectivity relation can be observed in Figure 4. The dengue fever starts with a very high viral burden (approximately $10^{5}$ to $10^{7}$ copies/mL in blood plasma) on day 0 of illness and grows sharply during the early days of the febrile period. Here, it should be noted that the variability of the within-host incubation period of dengue virus has been approximated by sampling day 0 viraemia level. During the early fever days, the transmission probability is high enough to infect upto 90% of the mosquito population. However, the virus assisted transmission of DI particles ($I_{VD}$) persists for longer, even after the virus is mostly cleared, while the only DI particles transmission is much lower than the standard virus and the assisted transmission. We want to explain the dynamics in terms of the replication competition in the within-host and within-vector dynamics. In the very early days of illness, the host plasma viraemia is mostly populated by the standard virus and they replicate and transmit through mosquito bites. During these days, the population of virus-infected host cells grows faster and releases virus. The DI particles and cells infected by both the standard and DI particles appear to grow with a delay [32]. That delay is also reflected here in terms of transmission. The $I_{VD}$ mosquito population starts to reach high values as soon as the $I_{V}$ starts to fall.

As we have discussed, the primary and secondary infections in the within-patient models can be classified with the help of their viraemia time profiles. The same situation is also reflected here in the mosquito model. As the majority of the DENV-2 and DENV-3 infected patients were diagnosed with a secondary infection, the POMs for DENV-2 and DENV-3 in the current model show sharper growth in transmission than DENV-1 and DENV-4. In contrast, the $I_{VD}$ transmission for DENV-2 and DENV-3 are lower than those in the case of DENV-1 and DENV-4. Figure 2 shows the density distribution of the $I_{VD}$ trajectories for different serotypes. These results suggest that the more secondary infections occur, the more the standard virus will be transmitted with respect to DI particles.

The controlled profiles of the infection fractions of the mosquito population show log-scale reduction from the uncontrolled profiles. The maximum reduction is observed in the profiles of $I_{V}$ after applying the control ($I_{VC}$), whereas the $I_{VD}$ profiles are not reduced completely ($I_{VDC}$). These results explain the underlying trade-off between the transmission of virus and DI particles that co-transmission ($I_{VD}$) is necessary for significant transmission of DI particles ($I_{D}$). The goal of the DI particle mediated intervention is to completely block the virus transmission with a persistent good level of DI particles transmission. However, the passage of DI particles is not possible without the assistance of the transmission of a helper virus. Hence, a high-efficiency passage of DI particles needs to allow a lower minimum level of virus to be transmitted as co-transmission. Nevertheless, the allowed level of viral transmission is not enough to produce dengue fever or endemic outbreak.

Although quantitative detection of the existence of viral load in different mosquito body parts is a common experiment, it is hard to detect if there exists any DI RNA. To tackle such problems, a predictive model may address a number of questions. We believe this model is the first to explore the possible scenario of a mosquito population, infected by blood-feeding themselves from dengue infected human host. We hope to develop another model on virus replication and the mechanism of natural occurrence of DI particles via genome deletion and mutation. This model will consider the multi-class queue of (+)ssRNA (full length and defective) and investigate the delay in terms of dengue control strategy.

The authors declare that the data supporting the findings of this study are available within the paper.

## Figures and Tables

**Figure 1 viruses-12-00558-f001:**
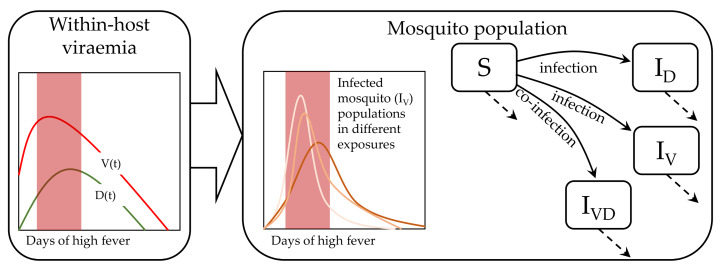
Dengue virus host-to-vector transmission model: The virus (V(t)) and the defective interfering (DI) particles (D(t)) are transmitted from infected hosts to mosquitoes. The susceptible mosquitoes (*S*) can be infected by virus or DI particles to generate the two types of infected mosquitoes, $I_{V}$ and $I_{D}$. Further, a dually-infected population ($I_{VD}$) is generated from co-infection by D(t) and V(t), simultaneously. The model has been considered in two-compartments. The D(t) and V(t) are generated in the within-host compartment and infect the mosquito population in the other compartment. For each within-host model (shown in left box), multiple plausible transmission models has been calibrated (shown by multiple lines). The shaded domain indicates the days of high fever.

**Figure 2 viruses-12-00558-f002:**
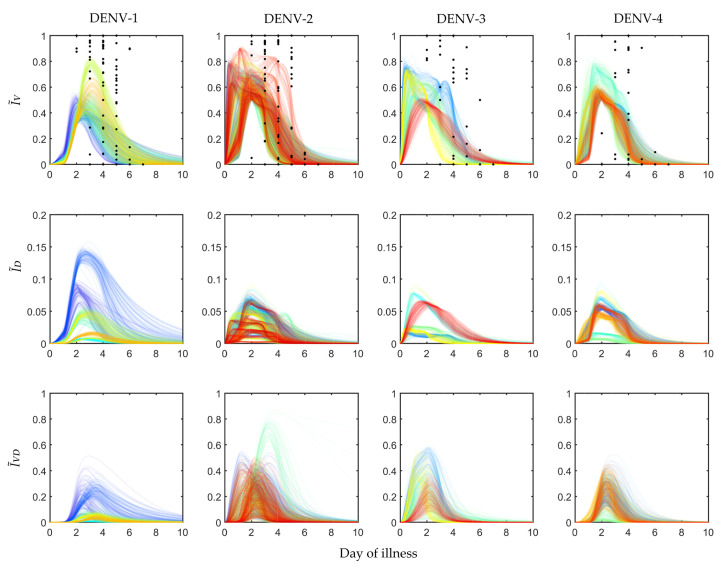
Population of models for infected mosquitoes: The dengue virus infected mosquito data for 408 exposure experiments with 208 hospitalised dengue patients reported in Nguyen et al. [33] are calibrated to construct serotype-specific population of models (POMs). The black scattered points in the top panel represent the fraction of viral infected mosquitoes (${\tilde{I}}_{V}$) observed in the exposure experiment. The group of lines in different colours (in the top, middle and bottom panels) are the calibrated POMs outputs for different patient models in the population.

**Figure 3 viruses-12-00558-f003:**
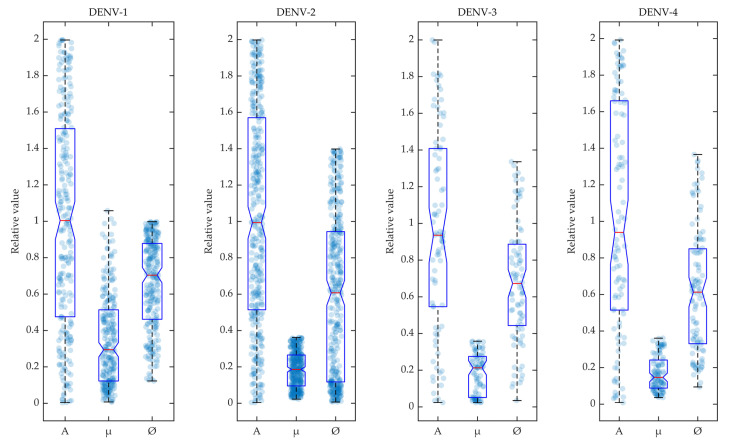
Variability in accepted model parameters: The parameters accepted in the POMs are shown in box plots for the four serotypes. The density of the parameters (*A*, μ, ϕ) are shown in the scatter plots in the background of the box plots.

**Figure 4 viruses-12-00558-f004:**
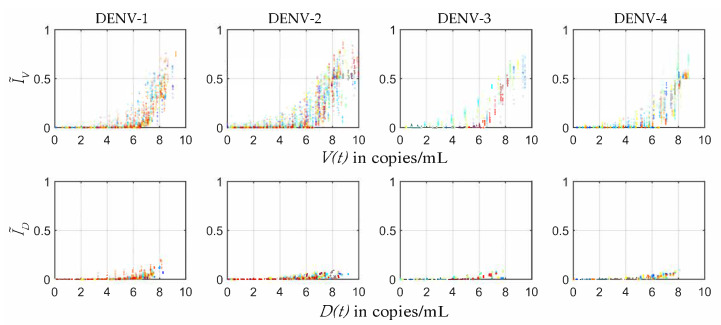
Mosquito infectious dose: The infected mosquitoes by virus (${\tilde{I}}_{V}$) and DI particles (${\tilde{I}}_{D}$) are shown in scattered plots with respect to the ${\mathrm{Log}}_{10}$ values of corresponding within-host virus (V(t)) and DI particles (D(t)) levels. Different colours represent different patient models. These scatter plots represent the phase portrait of the data shown in Figure 2 versus the viraemia data reported in our previous model [32] and is a way to estimate the ranges of different mosquito infectious doses (MIDs), say $MID_{50}$.

**Figure 5 viruses-12-00558-f005:**
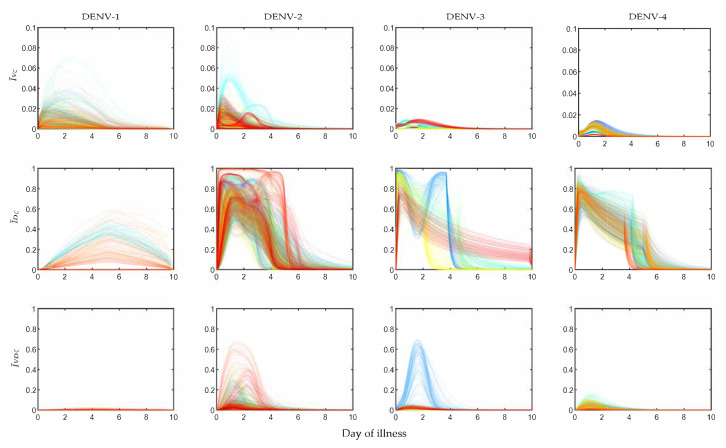
Controlled population of transmission models: Each serotype-specific POMs (from Figure 2) are considered after applying optimal bang-bang control to construct the controlled POMs (cPOMs). The infected mosquitoes by virus only (${\tilde{I}}_{VC}$), by DI only (${\tilde{I}}_{DC}$) and by both (${\tilde{I}}_{VDC}$), are computed by using the controlled within-host plasma viraemias in the populations for four dengue serotypes.

**Figure 6 viruses-12-00558-f006:**
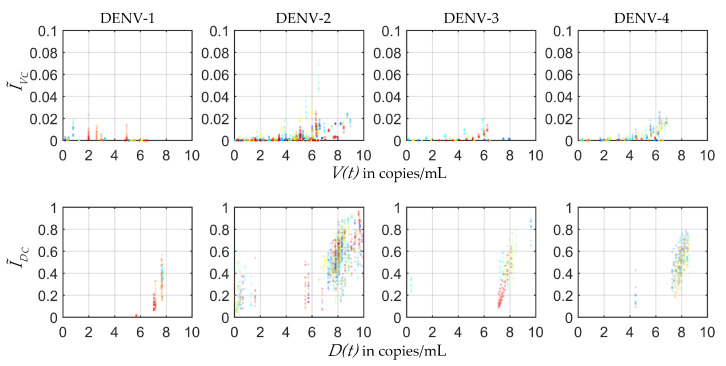
Controlled MIDs: Scatter plots show the controlled infected fractions ($I_{VC}$ and $I_{DC}$) of the mosquito population versus the controlled patient viraemias (V(t) and D(t)). The transmission of DI particles with respect to the ${\mathrm{Log}}_{10}$ values of the plasma DI particles (D(t)) has notable rise after applying the control (see Figure 4).

**Figure 7 viruses-12-00558-f007:**
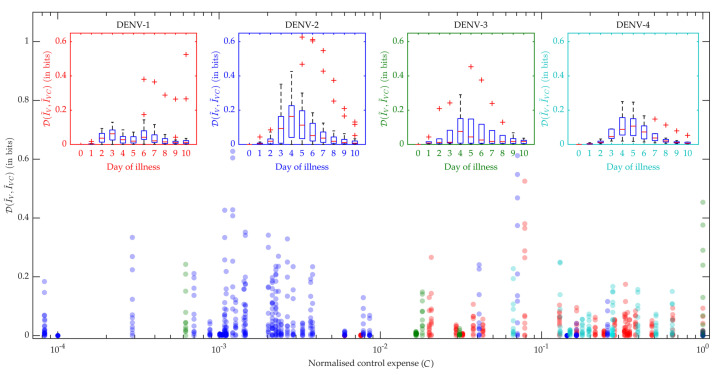
Control efficiency: The reduction in the human-to-mosquito virus transmission was evaluated by Jensen-Shannon (J-S) divergence (D), calculated between the distributions of the virus infected mosquitoes before (${\tilde{I}}_{V}$) and after (${\tilde{I}}_{VC}$) applying the optimal control. The box plot for each serotype explains the variation in the J-S divergence on each day of the febrile period (in the insets). The main scatter plot compares the efficiency of the applied optimal controls for different serotypes, in terms of J-S divergence versus normalised control expense (*C*) (red: DENV-1, blue: DENV-2, green: DENV-3, cyan: DENV-4). The control expense was computed by the area under the control curve [32].

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
