# Peer review of "Administration of Defective Virus Inhibits Dengue Transmission into Mosquitoes"

_viruses, 2020, doi:10.3390/v12050558_

Round 1

Reviewer 1 Report

Remarks to Authors:

This paper explores the impact of defective interfering particles on limiting dengue transmission from human to mosquito by applying a within-host dengue model to a mosquito model, relying on clinical experimental data relating human viral load to mosquito infectiousness. The paper tackles an interesting question. However, descriptions of how data is used to estimate model parameters is not well-described. The impact of model assumptions on model results need to be more clearly described. Most importantly, the implications of how he model results relate to public health control interventions need to be described.

Major comments:

Introduction:

In the last paragraph of the Introduction, what is missing is a clear connection of how the published DI models will be applied to understand how DI particles can be used to limit transmission from hosts to vectors. I would also include in the Introduction a brief description of the data used to calibrate the models that link human viral load to mosquito infectivity.

Methods:

Generally, more information is needed in Methods, and any Methods described in the Results that are not in the Methods should be moved to the Methods.

Are primary and secondary infections ever modeled differently? What about asymptomatic versus symptomatic infections?

How are the simulation models calibrated based on the Nguyen paper? Are any parameters actually estimated or is the output of the model in Figure 2 simply plotted alongside the Nguyen data?

It seems you are assuming a fixed rate of mosquito infection (f) that is not time-dependent or dependent on the amount of virus in the human. How is this being quantified from the Ngyuen et al paper? How would the results change if this rate depended on human viral load directly?

More description is needed of how the parameters in the mosquito model are being sampled using LHS. The range for parameters are given in the Results (should be in Methods), but what distribution is assumed when applying LHS? Do you assume each parameter is sampled from a uniform distribution?

A description of how the MID50 is estimated is needed in the Methods.

Description of the Jensen-Shannon Divergence should be moved to the Methods. Please define the normalized control expense. I would also use a different parameter than A since this is used before in the transmission model.

Results:

It would be helpful to more directly show how the degree of reduction in human viral load due to DIs results in a reduction in the MID50.

More text or simulation results that show the impact of coinfection on model results would be helpful. The Results section on coinfection are not well understood based on the current figures in the paper.

Generally, results should more clearly demonstrate how public health interventions (ie the number and timing of administrating DI particles) impact the reduction in the number of infectious mosquitos. There are many plots shown but I don’t have a good sense of how to directly relate the administration of DI particles to the reduced number of infected mosquitos. For example, instead of applying the Jensen-Shannon Divergence to the normalized control expense, show the Jensen-Shannon Divergence as the number of DI particles needed to be administered per human, or another more relevant variable.

Discussion:

Please describe how the model assumptions impact the conclusions of the paper.

Please also describe how the DIs can be applied in a real-world setting. Patients only go to the hospital is they are highly symptomatic, even though we know the majority of dengue infections are asymptomatic. How would this impact the application of DIs?

Is there any example of how DIs have been implemented for other viruses? How do these relate to dengue?

Minor comments:

Introduction:

“Although four serotypes have different interactions with the host antibodies, they result the same disease with the same clinical symptoms.” It is known that certain serotypes (ie DENV-2) typically cause more severe disease than other serotypes. I would either mention variability in clinical manifestation based on serotype or take out this sentence.

Clarify “We suggest this phenomenon as a potentially efficient dengue control.” This is too vague. Suggested chance: “The mechanisms underlying the transmission of virus from human to mosquito can be exploited as a potential effective dengue control.”

“In consequence, the global spread of dengue is pandemic, accentuated, for example, through increased travel between countries and urban growth.” This sentence is unclear. There are grammatical errors and I am not sure what is trying to be said.

The paragraph on vaccination is too broad. The main reason that developing a vaccine for dengue is difficult is because of cross-reactivity between serotypes. The Sanofi vaccine and other vaccines in development should be cited here.

Reviewer 2 Report

This is an interesting work, tracking the population dynamics of viruses and their DI particles from human hosts to mosquito vectors, seeking control conditions to drive virus levels in vectors down and thereby halt transmission. Strengths are the use of data from human-to-mosquito transfer to inform the model.

Weaknesses are lack of details on terminology for dengue and mosquito dynamics (detailed below). A further weakness is discussion about the potential practical application of the strategy. Specifically, it is not clear what information or measurements would be needed to implement the optimal control strategy - does one need to know the relative levels of virus and DI particles in the host to know whether more DI particles should be added? Further comments follow below.  

p.3 last paragraph: containing neucleotide deletions, <- fix typo on "nucleotide"  

p.5 Methods. In practice, what does "adding excess DI particles" mean? Don't they need to be delivered to the site of infection in the host? How would this be done in practice?  

p. 6 What are "high fever days" for dengue infections and why are they relevant for the mosquito exposure experiment? What is a "mosquito pool?" A population of mosquitos, a pool of water that mosquitoes share, or something else? Please define your terms so non-dengue and non-mosquito experts can understand the most relevant points of your work.

Does "triggered immune response" refer to innate or adaptive immunity or both? How is it relevant to the modeling? 

Figure 1. How are I_V different from I_D infected mosquitoes? It is not clear why a susceptible mosquito that picks up both virus and DI particles from a host is not immediately I_VD. The main text indicates that "while taking a blood meal the mosquito can be infected in three different ways, by the virus, by the DI particles or dually." But Fig. 1 suggests the dual infected state must arise from either of the singly infected states; there is no direct arrow from S to I_VD, as would be implied by the main text.

Explain or provide the minimal relevant points about what the "optimal bang-bang control strategy for patient viraemia reduction" entails. Summarize for the readers so they do not need to seek out this information from ref. 31. 

p.8 How is it known that mosquitos die naturally before they can recover from dengue virus infection? Please cite relevant literature.  

p. 8-9. Are ODE models valid? They appear to assume levels of V or D in a blood meal are sufficiently high that variation from intrinsic stochasticity can be ignored. This possibility is not mentioned among the sources of underlying variability. Please include some discussion of why this factor can be assumed negligible.  

p.11 Does the solution to the optimal control problem require that V and/or D be measured? If no, how does one know what would be the optimal strategy? If yes, how would this strategy be implemented in practice? Specifically, how would one quantify the level of D?  

p. 22 What does a very high viral burden of about 10^5 to 10^7 mean? Is this a number of virus particles per human host, per ml of blood or something else. Given the load level and the volume to blood sampled by a mosquito, what is the typical level of virus (number of infectious particles) transferred to a mosquito during a blood meal?  

General comment: In classical studies of virus-DI particle dynamics chaotic (in time) or oscillatory behavior in time or space (refs 29 and 30) are most commonly observed, reflecting features of the predator-prey relationship between DI and virus particles, respectively. None of that classical behavior seems to play out in the current model involving co-transmission of these particles. Why not? A clear explanation would be helpful for the readers familiar with classical DI particle dynamics.

Round 2

Reviewer 2 Report

Authors have been responsive to questions and suggestions.